# Innovative Approaches to Optimize Clinical Transporter Drug–Drug Interaction Studies

**DOI:** 10.3390/pharmaceutics16080992

**Published:** 2024-07-26

**Authors:** Sabina Paglialunga, Natacha Benrimoh, Aernout van Haarst

**Affiliations:** 1Scientific Affairs, Celerion, Tempe, AZ 85283, USA; 2Data Management and Biometrics, Celerion, Montreal, QC H4M 2N8, Canada; 3Scientific Affairs, Celerion, Belfast BT9 6AD, UK

**Keywords:** breast cancer resistance protein (BCRP), cocktail drugs, drug–drug interaction (DDI), endogenous biomarkers, multidrug and toxin extrusion protein (MATE), organic anion transporter (OAT), organic anion-transporting polypeptide B (OATP1B), organic cation transporter (OCT), P-glycoprotein (P-gp)

## Abstract

Of the 450 cell membrane transporters responsible for shuttling substrates, nutrients, hormones, neurotransmitters, antioxidants, and signaling molecules, approximately nine are associated with clinically relevant drug–drug interactions (DDIs) due to their role in drug and metabolite transport. Therefore, a clinical study evaluating potential transporter DDIs is recommended if an investigational product is intestinally absorbed, undergoes renal or hepatic elimination, or is suspected to either be a transporter substrate or perpetrator. However, many of the transporter substrates and inhibitors administered during a DDI study also affect cytochrome P450 (CYP) activity, which can complicate data interpretation. To overcome these challenges, the assessment of endogenous biomarkers can help elucidate the mechanism of complex DDIs when multiple transporters or CYPs may be involved. This perspective article will highlight how creative study designs are currently being utilized to address complex transporter DDIs and the role of physiology-based -pharmacokinetic (PBPK) models can play.

## 1. Introduction

The two classes of cell membrane transporters associated with drug transport are adenosine triphosphate (ATP)-binding cassette (ABC) and solute carrier (SLC) membrane transporters. Members of these transporters are responsible for shuttling drugs between the gut, the systemic circulation, and organs/tissues, such as the liver and the kidneys, and thus play an integral role in drug absorption, distribution, metabolism, and excretion. ABC membrane transporters such as P-glycoprotein (P-gp) and breast cancer-resistant protein (BCRP) utilize ATP hydrolysis to function as ‘efflux’ transporters, transferring drugs and endogenous solutes out of enterocytes or hepatocytes into the gut lumen or bile, respectively (review in [1]). On the other hand, SLC membrane transporters such as organic anion transporters (OATs), organic anion-transporting polypeptides (OATPs), and organic cation transporters (OCTs) are generally considered ‘influx’ transporters, responsible for the movement of drugs and endogenous solutes across membranes into enterocytes, hepatocytes, endothelial, and renal cells (Table 1).

## 2. Clinical Transporter Drug–Drug Interaction Studies

Transporter inhibition due to the interaction of co-administered drugs, genetic polymorphism, or even diseased states can affect patient safety or lead to reduced drug efficacy. Therefore, a drug–drug interaction (DDI) study is a critical aspect of drug development to ascertain if an investigational drug is a transporter substrate or perpetrator. The assessment of a study drug’s DDI risk potential starts during nonclinical development with a suite of in vitro assessments, often prior to initiating clinical trials and depending upon anticipated administration and excretion routes as well as toxicity profiles. However, the categorical separation between the effects of study drugs as substrates or their inhibitory effects in in vitro transporter assays can, in some cases, be difficult to interpret. For instance, the false positive rate of OATP1B1/3 inhibition in in vitro assays for predicting clinical findings has been reported to be up to 33% [5]. Hence, clinical DDI studies are generally inevitable to confirm DDI risk potential in humans, in which there are nine main transporters of clinical relevance as highlighted in Table 1.

The Food and Drug Administration (FDA) provides a list of transporter substrates and inhibitors for clinical DDI studies [2]; however, not all the products are suitable or tolerable for healthy volunteer administration and none of these are considered ‘clinical index drugs’, unlike with cytochrome P450 (CYP) enzymes for DDI studies. Per the FDA, a clinical index drug predictably affects a (distinct) metabolic pathway. However, many of the drug transporter substrates and inhibitors listed in Table 1 also engage with CYP enzymes or other transporters, making the interpretation of results challenging. For instance, itraconazole is a P-gp inhibitor, but also a strong inhibitor of CYP3A; thus, the P-gp effects of an investigational drug co-administered with itraconazole can be confounded if the drug in question is metabolized by this CYP enzyme. Moreover, itraconazole is also recognized as a BCRP inhibitor, further complicating result interpretation [6]. Multiple DDI studies may be needed, in addition to nonclinical assays and physiology-based pharmacokinetic (PBPK) modeling to ‘tweeze’ out transporter-CYP effects as illustrated in the case of darolutamide, a novel androgen receptor antagonist [7]. In in vitro studies and three separate clinical DDI studies, combined with population pharmacokinetic (PK) modeling, Zurth et al. evaluated the effects of itraconazole (a CYP3A4, P-gp, and BCRP inhibitor; NCT03048110) on darolutamide and of darolutamide on dabigatran etexilate (a P-gp substrate; NCT03237416) and rosuvastatin (a substrate for BCRP, OATP1B1/B3, and OAT3; NCT02671097) to demonstrate that darolutamide is both a substrate and strong inhibitor of BCRP as well as minor inhibitor of OATPB1/B3 [7]. 

To potentially reduce the number of clinical trials, assist with the interpretation of complex transporter DDI results, and guide the adjustment of therapeutic dose levels in clinical practice, strategic approaches such as the use of drug cocktails, assessment of endogenous drug transporter biomarkers, and application of PBPK models can be leveraged [8,9,10].

## 3. Cocktail Drug Approach for Transporter DDI Studies 

The cocktail probe approach combines multiple well-characterized substrates to evaluate the inhibition or induction potential of a study drug for different CYP enzymes or transporters in one study. The mixture of substrates should be validated, showing no drug interactions (e.g., no PK differences when dosed alone vs. in a cocktail), specific for the individual CYP enzyme or transporter, and the study must be sufficiently powered with adequate sample size based on study objectives. While dozens of drug cocktails have been validated for CYP enzymes [11], only a handful exist solely for transporters (Table 2). 

From a historical perspective, Martin et al. first demonstrated that there was no interaction between digoxin and rosuvastatin in 2002, laying the foundation for these drugs to be combined as a P-gp and BCRP/OATP1B1/3 substrate cocktail [12]. Building off this work, in 2016 Stopfer et al. explored the first transporter cocktail consisting of digoxin (0.25 mg), furosemide (5 mg), metformin (500 mg), and rosuvastatin (10 mg) to assess P-gp, OAT1/3, OCT2/MATE1/2-K, and BCRP/OATP1B1/3, respectively [13]. PK analysis revealed that digoxin and metformin exposures were equivalent when administered in the cocktail form compared to the substrate alone, with the geometric mean ratios for the 90% confidence intervals (CIs) of the area under the concentration–time curve (AUC) and maximal concentration (C_max_) between 80 and 125%. A small decrease in furosemide C_max_ (but not AUC) was observed, which, however, did not affect the sensitivity of furosemide as a probe. In addition, rosuvastatin exposure increased by nearly 40% when administered as part of the cocktail compared to when dosed alone. The authors speculated that metformin may be the perpetrator associated with the increased rosuvastatin exposure, but also stressed that the effect was minimal, as rifampin, a strong OATP1B inhibitor, typically increases rosuvastatin AUC by 5-fold [13]. By adjusting the substrate doses, the ‘Boehringer’ cocktail was subsequently optimized and validated for use in DDI studies; digoxin (0.25 mg), furosemide (1 mg), metformin (10 mg), and rosuvastatin (10 mg) revealed no mutual PK interactions and had a favorable safety profile in healthy volunteers in multiple trials [14,15].

**Table 2 pharmaceutics-16-00992-t002:** Validated and exploratory transporter cocktails.

Cocktail	P-gp	BCRP	OATP1B1	OATP1B3	OAT1	OAT3	OCT2/MATE1/2-K
**Validated Cocktails**							
P-gp/BCRP Martin et al. (2002) [12]	Digoxin 0.25 mg	Rosuvastatin 10 mg					
“Boehringer” Wiebe et al. (2020) [15]	Digoxin 0.25 mg	Rosuvastatin 10 mg			Furosemide 1 mg		Metformin 10 mg
“Cologne” Trueck et al. (2019) [16]	Digoxin 0.5 mg	-	Pitavastatin 2 mg		Adefovir 10 mg	Sitagliptin 100 mg	Metformin 500 mg
“Merck Microdose” Prueksaritanont et al. (2017) [17]	Dabigatran 375 µg	-	Pitavastatin 10 µg		-	-	-
	-	Rosuvastatin 25 µg					
	Atorvastatin 50 µg						
**Exploratory Cocktail**							
Ogasawara et al. (2021) [18]	Digoxin 0.25 mg	Rosuvastatin 10 mg			-	-	Metformin 1000 mg

Next, Trueck et al. proposed a 5-probe cocktail consisting of adefovir (10 mg), sitagliptin (100 mg), metformin (500 mg), pitavastatin (2 mg), and digoxin (0.5 mg) to further distinguish transporter effects [16]. Adefovir and sitagliptin were selected to discriminate between OAT1 and OAT3 [19,20], while pitavastatin is sensitive to OATP1B1 and OATP1B3 [21]. Overall, the ‘Cologne’ cocktail combination was found to be safe in healthy subjects and no major mutual PK interactions were observed; only the upper 90% CI for adefovir exposure was slightly above the 125% threshold [16]. 

Interestingly, Prueksaritanont et al. took a more conservative approach and validated a cocktail that administers microdose levels of midazolam (10 µg), dabigatran etexilate (375 µg), pitavastatin (10 µg), rosuvastatin (25 µg), and atorvastatin (50 µg), and evaluated the inhibitory effects of rifampin, itraconazole, and clarithromycin as perpetrator drugs [17]. The combination of drug substrates allowed to differentiate between the effects on P-gp, OATP1B, and BCRP (Table 2), because the substrates have different sensitivities for these transporters, with pitavastatin displaying greater selectivity as a substrate for OATP1B than rosuvastatin. In this cocktail combination, dabigatran etexilate is the preferred P-gp substrate over digoxin. Digoxin has a narrow therapeutic window, exhibits low P-gp sensitivity due to high oral bioavailability (60–80%), and thus may not adequately capture a ‘worst-case’ victim DDI potential caused by intestinal P-gp [22]. However, it should be noted that, because dabigatran etexilate is a prodrug of the P-gp substrate dabigatran, the cocktail is not suitable for the evaluation of systemic P-gp activity. Moreover, at a microdose level, dabigatran etexilate may also be a substrate for CYP3A [23]. Although the use of microdoses was fully validated in the above case, it is important to underscore that the application of microdose levels of substrates, in general, may not necessarily allow the translation of the observed effects to the therapeutic dose level (e.g., due to the non-linearity of PK). This is also highlighted in the FDA’s DDI guidance [24]. One general consideration for the cocktails listed above is that most ‘statin’ drugs are also considered CYP substrates and this may need to be evaluated closely when interpreting results. 

While there are only a few validated transporter cocktails (Table 2), others have applied this approach in an experimental manner to examine the potential effect of an investigational product on transporter inhibition. For example, Ogasawara et al. combined digoxin (0.25 mg), rosuvastatin (10 mg), and metformin (1000 mg) as a drug cocktail to interrogate the effect of fedratinib, an oral selective Janus kinase 2 inhibitor, on drug transporters [18]. Nonclinical studies demonstrated that fedratinib inhibited P-gp, BCRP, OATP1B1/B3, and OCT2/MATE1/2-K. The clinical study revealed that plasma exposures to digoxin and rosuvastatin were generally comparable in the presence or absence of fedratinib, suggesting no clinically significant interactions with these transporters. It should be noted that the metformin dose selected in this ‘exploratory’ cocktail was not validated as per the examples above but was chosen since it is a therapeutic dose often prescribed to patients, and therefore does not rule out or account for a potential metformin–rosuvastatin interaction. Interestingly, metformin renal clearance was reduced by 36% when co-administered with fedratinib and this resulted in elevated glucose levels during an oral glucose tolerance test, suggesting a potential inhibitory interaction with OCT2 and MATE transporters. However, the effect was deemed minor, as it is not discussed in the Drug Interaction section of the drug label [25]. This example illustrates how an experimental cocktail approach can be insightful, although caution may need to be taken when interpreting results as the cocktail is not validated for a particular context of use. 

## 4. Role of Endogenous Biomarkers in Transporter DDI Studies

Another approach to monitor changes in transporter function upon the administration of a novel study drug is to measure endogenous solute concentrations as ‘biomarkers’ of transporter activity. To be considered a suitable biomarker, the biosynthesis and metabolism of the endogenous product should be well established and display good selectivity, specificity, and sensitivity for the transporter in question. One such set of biomarkers that has been gaining attention by drug developers [26] and regulators [27] are coproporphyrins I (CP-I) and III (CP-III), which are heme metabolites that are both taken up by OATP1B1 and B3 (Table 1). Similar to the effects seen for the OATP1B1 probe rosuvastatin, CP-I plasma concentrations transiently increase following administration of 600 mg rifampin, a strong OATP1B inhibitor, reaching a peak after ~4 h [28]. Moreover, the AUC ratio (AUC_inhibitor_/AUC_control_) of CP-1 rose from 3.0 to 4.6 with increasing single doses of rifampin (300 and 600 mg) [26]. In addition, Mori et al. demonstrated dose-dependent increases in CP-I concentrations with the ascending administration of oral rifampin (150 to 600 mg), a strong OATP1B1 inhibitor [29]. Their analysis revealed excellent sensitivity of CP-I to discriminate dose-dependent OATP1B1 effects with minimal inter-day and diurnal variation, making CP-I an ideal biomarker [29]. For a comprehensive discussion on the role of endogenous transporter biomarkers, we refer readers to a recent review article by Arya et al. [30], also featured in this Pharmaceutics Special Issue.

In a clinical study involving fenebrutinib, a Bruton’s tyrosine kinase inhibitor, both CP-I and CP-III were evaluated. Nonclinical data suggested that fenebrutinib potentially inhibited BCRP and/or OATP1B1 [31]; therefore, these biomarkers were measured during a DDI study with rosuvastatin to determine the OATP1B1 contribution. As anticipated, rosuvastatin plasma exposure was increased by more than 2.5-fold with fenebrutinib co-administration compared to rosuvastatin treatment alone; however, CP-I and CP-III plasma concentrations were similar in both conditions, suggesting that fenebrutinib inhibits BCRP activity, yet not OATP1B1 [31]. These results corroborated PBPK prediction and informed concomitant medication recommendations in later phase studies [32]. 

Comparing the performance of CP-I vs. CP-III, Kalluri et al. found that CP-I levels increased with greater glecaprevir exposure, an OATP1B1/3 inhibitor, whereas there was minimal change in CP-III concentrations [33]. In addition, the overall low concentrations of CP-III were near the lower limit of quantification, complicating the analysis. Therefore, the authors concluded that CP-I is a more sensitive and robust biomarker that can help inform OATP1B1/3 inhibition potential in early clinical development [33]. This biomarker approach to evaluate DDI risk assessment is supported by regulatory agencies as an emerging alternative to a dedicated trial when monitored in early phase studies, as described in the final ICH M12 guidance [6]. Per the guidance, if the ratio of post-dose to baseline CP-I peak concentration (C_max_) and total exposure (AUC) are less than 1.25, then a low likelihood of a clinical DDI via OATP1B inhibition is anticipated. 

There are several other endogenous transporter biomarkers currently being applied in clinical trials (Table 1). For example, N1-methylnicotinamide (also known as NMN), derived from tryptophan and vitamin B3 metabolism, is actively transported by OCT2 and MATEs into urine relatively unchanged, making it a suitable biomarker of OCT2 and MATE1/2K activity [34,35,36]. As the interest and utility of endogenous transporter biomarkers grow, advancements in this area include multiplexed bioanalysis to quantify several transporter biomarkers simultaneously using chromatography tandem mass spectroscopy methodology [37]. Moreover, investigation into novel biomarkers reflecting P-gp and BCRP activity is also underway. Jin et al. identified azelaic acid ( AzA) as a putative endogenous substrate of both OATP1B3 and P-gp, and suggested that changes in AzA may reflect alterations in the directional transport from blood to bile via OATP1B3 and P-gp [38]. In addition, Shen et al. are evaluating riboflavin as a potential BCPR biomarker [39]. Building on the growing interest in endogenous biomarkers for SLC transporters, Rodrigues proposes a reimagined framework to de-risk DDI and preclude the need for a clinical study when warranted [40]. This proposed approach includes a dynamic prediction of the percent inhibition for each SLC transporter based on biomarker exposures (i.e., AUC ratio) and renal clearance rates obtained during phase I studies as well as biomarker PBPK modeling.

## 5. PBPK Modeling

PBPK modeling is a mechanistic dynamic tool gaining traction in drug development as it leverages multi-compartmental models representing organs and blood flow to simulate clinical exposure data and assess untested clinical scenarios such as DDIs. Since 2017, an estimated two-thirds of the PBPK publications evaluated DDIs, highlighting the importance of this tool for DDI risk potential [41]. PBPK models have been used to estimate the clinical DDI magnitude and assist in designing clinical DDI studies and have even been applied in lieu of conducting a clinical trial. Ibrutinib, a tyrosine kinase inhibitor approved for the treatment of lymphoma, was the first drug to receive FDA acceptance with a PBPK modeling approach in 2013, in which 24 label claims were supported by modeling rather than clinical studies [42]. There are now several examples where regulatory agencies have accepted PBPK modeling data to inform drug labeling (see recent review articles [8,43,44]). While there are fewer commercially available validated PBPK models for transporter- than CYP-mediated DDIs, this is in part due to the complex nature of efflux and uptake transporter kinetics as well as quantification of absolute transporter expression [41]. Nonetheless, transporter PBPK models have had a ‘high impact’ on drug development and regulatory decisions. Taskar et al. reviewed over two dozen examples of transporter-mediated DDI PBPK analyses from new drug applications and published studies, and found several instances where the models were sufficient to waive clinical DDI studies [43]. Such is the case for mobocertinib, a kinase inhibitor approved for lung cancer, where in vitro and clinical data were used to build and validate a PBPK model to assess an interaction with P-gp. The drug label states that no clinically meaningful difference in digoxin or dabigatran etexilate (P-gp substrates) “are predicted” when co-administered with multiple doses of mobocertinib [45]. PBPK models can also be applied to elucidate complex transporter–CYP mediated DDIs as evaluated by Bowman et al. for pralsetinib, a tyrosine kinase inhibitor [46]. Here, the authors developed and validated a PBPK model for pralsetinib to delineate the CYP3A vs. P-gp contribution for a victim drug that can be applied as a framework for further model applications. Finally, PBPK modeling can be combined with endogenous biomarkers to provide a deeper understanding of the mechanisms of transporter inhibition. For example, Yoshikado and colleagues developed and validated a PBPK model for CP-I to evaluate potential OATP1B inhibitors [47]. The authors suggest that this analysis could be applied to predict the OATP1B DDI potential of a new chemical entity after inputting the CP-I values (i.e., in vivo inhibition constant, K_i_) obtained during a phase I dose-escalating study into the model. 

## 6. Transporter Gene Expression and Polymorphisms 

The gene expression of drug transporters constitutively varies across tissues, thus differentiating their roles in drug absorption, in the distribution of drugs to key organs such as the liver and the brain, and in the elimination of parent drugs and their metabolites (e.g., through excretion by the kidney). Furthermore, genetic variations may underlie structural inter-individual differences in the activity of drug transporter proteins [48]. Single-nucleotide polymorphisms (SNPs) have been described for multiple transporters and may have a significant impact on drug PK, safety, and efficacy [49]. For instance, the *ABCG2 c.421C>A* SNP is one of the most frequent polymorphisms in *ABCG2*, and is associated with 30–40% reductions in BCRP protein expression; this allele occurs across more than 30% of East Asian biogeographical groups, while is rare in African American, Afro-Caribbean, and Sub-Saharan African populations with an allele frequency of less than 5% [50]. Subjects that are heterozygotes for *ABCG2 c.421C>A* have substantially reduced BCRP activity, which has been associated with an increase of 80% in rosuvastatin exposure as compared to the exposure in subjects with a normal genotype [51]. Metformin efficacy may also be affected by genetic polymorphisms in OCT1 and 2 transporters and, as a result, patients with Type II Diabetes Mellitus (T2DM) may fail to establish adequate glycemic control [52]. While OCT1 may affect the PK of nearly 40% of prescription drugs [53], clinical DDIs via OCT1 have mainly been reported for metformin. However, OCT1 activity may vary considerably due to the genetic polymorphisms of *SLC22A1*, which may drastically affect drug PK [54]. Likewise, genetic variations in the *ABCB1* gene have been associated with differences in the efficacy of antidepressants, assumedly due to different levels of P-gp expression in the blood–brain barrier. As a consequence, antidepressants access to the brain is altered, which is supported by the assessment of lowered concentrations of antidepressant drugs in the brain in animal studies [55]. In another example, patients with certain *SLCO1B1* genotypes encoding for OATP1B1, polymorphisms such as *c.521T>C*, required reduced doses of statins to avoid statin-associated musculoskeletal symptoms [56]. In addition, genome-wide association analysis revealed significant gene–drug disposition effects for methotrexate (rs11045879; rs4149080) and ticagrelor (*c.521T>C*, rs4149056) with *SLOC1B1* SNPs (reviewed in Yee et al. [49]). Moreover, Yee et al. reported that differences in the OATP1B1 genotype may in fact modulate the effect of a perpetrator drug in a DDI study [57]. Therefore, it may be relevant to consider genotyping in clinical intervention studies, including DDI studies, so as to elucidate differences in PK and DDI effects retrospectively. 

Prospective genotyping can also be applied if a candidate drug has multiple routes of metabolism and elimination. Thus, a comparison of DDI effects in subjects with normal vs. decreased transporter function as a result of polymorphisms can help clarify the involvement of distinct clearance rates and elimination routes [6]. In the case that a particular transporter is assumed to be affected by a candidate drug yet an adequate probe substrate for that transporter is lacking, an alternative approach may be the enrolment of cohorts of subjects with different genotypes for that transporter. However, this will only be feasible if the distinct transporter polymorphisms result in substantially different transporter activities and also have sufficient prevalence.

## 7. Phenotypic Changes in Transporter Expression during Human Development and in Diseased States

Transporter protein expression and thus function can change in humans over time. For instance, liver OCT1 and P-gp content increase with age. However, ontogeny analysis revealed that the overall changes in transporter protein abundance during human development (i.e., fetus to infant to adult) are less pronounced than the several-fold differences observed in CYP enzyme expression over the same period [58]. Nonetheless, upregulated transporters can have clinically meaningful results. For example, renal OAT1, OCT2, and P-gp levels increase during pregnancy, resulting in greater amoxicillin, metformin, and digoxin renal clearance by ~50% or more (reviewed in Galetin et al. [48]). In addition, sex differences in transporter protein expression have also been noted, with gut and liver P-gp expression being slightly higher in males than females, but a definitive impact of such differences on drug PK has not been reported so far [48]. 

It is important to note that drug-induced changes in transporter function can be exacerbated in a diseased state. For example, patients with ischemia–reperfusion injury are at risk of developing acute kidney injury (AKI). AKI is characterized by a rapid decline in the glomerular filtration rate as well as the altered expression of several drug transporters in the basolateral and apical membranes of proximal tubular cells. The reduced expression of transporters such as OAT1 and OAT3 is thought to be a key culprit associated with reduced drug clearance in this condition (reviewed in Evers et al. [59]). Indeed, exposure to ciprofloxacin, a dual OAT1/3 substrate, increases in a dose-dependent fashion by 1.4- to 3.4-fold in patient cohorts with mild to severe kidney dysfunction compared to healthy controls, respectively [60]. In the above example, the impact of reduced renal transporter function can be evaluated in a clinical pharmacology renal impairment study. However, determining the effect of transporter engagement in other tissues is more complex. Positron emission tomography with a radiotracer can be applied to evaluate drug disposition and assess the impact of P-gp function [61]. This is an important consideration for drugs being developed to treat ischemic stroke, as imaging studies have demonstrated that blood–brain barrier permeability in human tissue rapidly increases after an ischemic stroke partly as a result of increased P-gp expression (reviewed in Evers et al. [59]). 

Transporters also play a role in multidrug resistance (MRD) diseases such as metastatic cancers. MRD is the resistance to multiple, structurally unrelated compounds, and is the major cause of chemotherapy failure. ABC membrane transporters such as P-gp are overexpressed in MRD cancer cells resulting in the efflux of chemotherapeutic agents, thereby reducing drug accumulation in tumor cells leading to drug resistance (reviewed in Gottesman et al. [62]). Several ‘generations’ of P-gp transporter inhibitors have been developed and added to chemotherapy regimens in the hopes of overcoming the MRD phenomena. Unfortunately, many failed due to poor inhibition and/or toxic side effects [63]; however, newer transporter inhibitors in development employing nanotechnology-based approaches have shown promising results [64].

## 8. Discussion

While the nonclinical evaluation of transporter-mediated interactions with in vitro assays is required as part of drug development [65], these results can be affected by high false positive rates and may not always reflect the clinical findings [5]. Therefore, clinical DDI studies can confirm if a study drug is a transporter substrate and/or perpetrator. However, there are now additional options to evaluate a candidate drug’s potential for drug interactions through transporters prior to conducting a clinical DDI study. For instance, if there is a sensitive endogenous biomarker for a particular transporter available, as in the case of OATP1B, employing such endogenous biomarkers in early-phase clinical studies (e.g., multiple ascending dose (MAD) studies) may be instrumental in determining the need of a standalone DDI study. Together with data from in vitro assays for a candidate drug using endogenous markers and probe substrates, the clinical findings from a MAD study can help build a PBPK model to further support the prediction of DDI potential [47]. Moreover, PBPK modeling can guide the design of clinical DDI studies or tease out the effect of transporter vs. CYP enzyme inhibitors and inducers, or potentially even eliminate the need for dedicated clinical DDI studies.

In recent years, newer substrates with greater selectivity for transporters have emerged, facilitating better interpretation of data from clinical DDI studies. These drug substrates can be applied alone or as a cocktail together with other substrates, and along with endogenous biomarkers. There are several advantages associated with a cocktail DDI study, such as reduced number of clinical studies and overall costs as a result of fewer subjects exposed to the study drug. In addition, this approach may also help to differentiate the contribution of different transporters in a single study, as shown with the ‘Merck Microdose’ cocktail [17], and other drug cocktails [15,16].

Apart from the main nine transporters the FDA recommends to evaluate for potential DDIs, other transporters may warrant clinical exploration. The rationale for these studies is typically driven by the investigational drug’s mechanism of action and/or (indirect) pharmacodynamic (PD) effects [66]. For instance, hepatitis B (HBV) and D (HDV) viruses use the bile acid transporter in the liver, sodium taurocholate co-transporting polypeptide (NTCP), to infect hepatic cells and, hence, novel antiviral drugs are designed to inhibit the binding of HBV and HDV to NTCP [67]. As a consequence, these antivirals may drastically increase bile acid levels in plasma [68] and thereby indirectly interfere with the PK and PD of co-administered drugs, warranting further investigations into the potential of DDIs. Sodium glucose co-transporter 2 (SGLT2) is another example of a membrane transporter of clinical relevance. SGLT2 inhibitors, such as empagliflozin and dapagliflozin, are common T2DM medications that reduce blood glucose levels through increased glucose urine excretion. Both inhibitors were evaluated in DDI studies with lobeglitazone, a peroxisome proliferator-activated receptor-γ agonist for a potential synergistic effect on glucose [69,70]. Moreover, drugs interacting with multidrug resistance-associated protein (MRP) also have a potential for clinically significant DDIs. In the kidneys, MRP2 serves as an efflux pump for endogenous substrates such as glucurodine and glutathione conjugates, but also xenobiotics like methotrexate, valsartan, and olmesartan. The efflux of these substrates through MRP can be inhibited by cyclosporine and efavirenz (reviewed in Veiga-Matos et al. [71]). These cases, however, represent a small fraction of the known drug–transporter interactions. Therefore, the understanding of an investigational product’s distribution, elimination, mechanism of action, and potential PD effects can guide the design of clinical studies assessing potential transporter DDIs.

## 9. Conclusions

Overall, strategic and innovative approaches such as cocktail studies, endogenous biomarker assessment, and PBPK modeling can be leveraged to evaluate the transporter DDI potential of an investigational drug, provide robust data, and streamline drug development. 

## Figures and Tables

**Table 1 pharmaceutics-16-00992-t001:** Efflux and influx drug transporter substrates, inhibitors, and endogenous biomarkers commonly applied in healthy volunteer clinical drug–drug interaction studies.

Transporter (Gene)	Tissue Expression *	Substrates **	Inhibitors **	Endogenous Biomarkers
**Clinically Relevant Efflux Transporters**				
BCRP (*ABCG2*)	Small intestine, colon, testis, blood–brain barrier, placenta, liver, and kidneys	Rosuvastatin	Cyclosporine Eltrombopag	None
P-gp (*ABCB1*)	Small intestine, liver, blood–brain barrier, and kidney	Dabigatran etexilate Digoxin Fexofenadine	Itraconazole Clarithromycin Quinidine Erythromycin Cobicistat	None
**Clinically Relevant Influx Transporters**				
OAT1 (*SLC22A6*)	Kidney, choroid plexus, liver, skeletal muscle, testes, and placenta	Adefovir Tenofovir Furosemide	Probenecid Gemfibrozil	Taurine Pyridoxic Acid HVA
OAT3 (*SLC22A8*)	Kidney, choroid plexus, testes, skeletal muscle, and adrenal glands	Famotidine Furosemide	Probenecid	6βHC GCDCA-S Pyridoxic Acid
OCT2 (*SLC22A2*)	Kidney, cortex and medulla, small intestine, and placenta	Metformin	Cimetidine Dolutegravir	NMN Creatinine Thiamine Tryptophan
OATP1B1 (*SLCO1B1*) OATP1B3 (*SLCO1B3*)	Liver and (placenta; OATP1B3)	Atorvastatin Fexofenadine Pitavastatin Pravastatin Repaglinide Rosuvastatin Simvastatin	Clarithromycin Cyclosporine Eltrombopag Gemfibrozil Rifampin (single dose) ***	CP-I CP-III Bilirubin Bile acids TDA HAD
MATE1 (*SLC47A1*) MATE2-K (*SLC47A2*)	Kidney, adrenal gland, and liver (skeletal muscle; MATE2-K)	Metformin	Cimetidine Dolutegravir	Creatinine Dopamine NMN Thiamine

* Expression profile as listed in UniProt.org database. ** Preferred drugs for healthy volunteer administration with a generally favorable safety profile [2]. *** Due to nitrosamine concerns, rifampin is currently not appropriate for healthy volunteer administration [3,4]. 6βHC, 6β-hydroxycortisol; ABC, ATP-binding cassette; BCRP, breast cancer resistance protein; CP, coproporphyrin; GCDCA-S, glycochenodeoxycholate-3-sulfate; NMN, N1-methylnicotinamide; HAD, hexadecanedioate; HVA, homovanillic acid; MATE, multidrug and toxin extrusion protein; OAT, organic anionic transporter; OATP1B, organic anion-transporting polypeptide B; OCT, organic cation transporter; P-gp, P-glycoprotein; SLC, solute carrier; TDA, tetradecanedioate.

## Data Availability

Not applicable.

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
