# Peer review of "Innovative Approaches to Optimize Clinical Transporter Drug–Drug Interaction Studies"

_pharmaceutics, 2024, doi:10.3390/pharmaceutics16080992_

Round 1

Reviewer 1 Report

Comments and Suggestions for Authors

Drug-drug interactions are a very important issue, and this review is an important paper that summarizes the issues at this stage and approaches to address them. The following are comments that the reviewers consider:

Page 7: The genetic mutations of transporters are described, for example, the authors described that 421C>A is the most frequent mutation of ABCG2. It would be better to have specific numbers such as frequency and population. Some mutations have racial differences; thus, please describe the differences in DDI due to racial differences.

Author Response

Comment 1: Drug-drug interactions are a very important issue, and this review is an important paper that summarizes the issues at this stage and approaches to address them. The following are comments that the reviewers consider:

Page 7: The genetic mutations of transporters are described, for example, the authors described that 421C>A is the most frequent mutation of ABCG2. It would be better to have specific numbers such as frequency and population. Some mutations have racial differences; thus, please describe the differences in DDI due to racial differences.

Response 1: Thank you for your comment, the manuscript was revised to include frequency and population metrics as well as clinical implications for key transporter polymorphisms for ABCG2 (see line 262-267) and OATP1B1 (see line 280-284).

Reviewer 2 Report

Comments and Suggestions for Authors

In their Communication “Innovative approaches to optimize clinical transporter drug-drug interaction studies” the authors presented several important topics regarding possible techniques for studying drug transporters. Overall, this article is carefully written and may be of interest for scientists looking for an overview about strategies for analyzing transport-mediated drug interactions. I have just some minor points to remark.

There are some minor imprecisions in some parts of the manuscript. In the abstract, the authors mentioned that membrane transporters are responsible for shuttling proteins. To the best of my knowledge, proteins are not described as substrates for transport proteins. In lane 119, they claimed that pitavastatin is sensitive for OATP1B1. However, pitavastatin has been described also as substrate for OATP1B3 (Hirano et al., 2004). In addition, in line 203, the authors mentioned an “OATP1B3-P-gp transport axis”. Since the substrate spectra of OATP1B3 and P-gp are different, this term is misleading and not generally accepted in the transporter community.

In chapter 6, the authors mentioned transporter gene expression and polymorphisms. This is an important topic also for transporter-mediated drug interactions and I suggest adding some more examples for established genetic variations with an impact on the pharmacokinetics and pharmacodynamics of drug substrates (e.g. OATP1B1*5). Furthermore, it is generally accepted to write genes and polymorphisms in italics.

Minor:

Table 1: Atorvastatin instead of Atrovastatin

Line 100: the mentioned reference refers to Stopfer et al. (2016) whereas in table 2 Wiebe et al. is mentioned.

Line 250: “transport protein” instead of “transporter molecule”

Author Response

Comment 1: In their Communication “Innovative approaches to optimize clinical transporter drug-drug interaction studies” the authors presented several important topics regarding possible techniques for studying drug transporters. Overall, this article is carefully written and may be of interest for scientists looking for an overview about strategies for analyzing transport-mediated drug interactions. I have just some minor points to remark.

Response 1: Thank you for your thoughtful comments and suggestions. We were pleased you found the strategic approaches useful for drug sponsors and scientists. Please find our responses to your comments below:

Comment 2: There are some minor imprecisions in some parts of the manuscript. In the abstract, the authors mentioned that membrane transporters are responsible for shuttling proteins. To the best of my knowledge, proteins are not described as substrates for transport proteins.

Response 2: Thank you for your comment, this has been corrected.

Comment 3: In lane 119, they claimed that pitavastatin is sensitive for OATP1B1. However, pitavastatin has been described also as substrate for OATP1B3 (Hirano et al., 2004).

Response 3: The text and table were updated to include pitavastatin as sensitive OATP1B1 and OATP1B3 substrates.

Comment 4: In addition, in line 203, the authors mentioned an “OATP1B3-P-gp transport axis”. Since the substrate spectra of OATP1B3 and P-gp are different, this term is misleading and not generally accepted in the transporter community.

Response 4: We have updated the sentence to clarify the OATP1B3 and P-gp “axis” as described in the Jin et al article. The revised text now states (line 212-214): “Jin et al. identified azelaic acid (a.k.a. AzA) a putative endogenous substrate of both OATP1B3 and P-gp, and suggested that changes in AzA may reflect alterations in the directional transport from blood to bile via OATP1B3 and P-gp”.

Comment 5: In chapter 6, the authors mentioned transporter gene expression and polymorphisms. This is an important topic also for transporter-mediated drug interactions and I suggest adding some more examples for established genetic variations with an impact on the pharmacokinetics and pharmacodynamics of drug substrates (e.g. OATP1B1*5). Furthermore, it is generally accepted to write genes and polymorphisms in italics.

Response 5: In line with the Reviewer’s recommendation, we have added several examples of how drug PK can be impacted by transporter polymorphisms. Moreover, we have consistently used italics when referring to specific gene polymorphisms.

Minor Comments:

  • Table 1: Atorvastatin instead of Atrovastatin
  • Line 100: the mentioned reference refers to Stopfer et al. (2016) whereas in table 2 Wiebe et al. is mentioned.
  • Line 250: “transport protein” instead of “transporter molecule"

Response to minor comments: All minor comments have been addressed in the revised version.

Reviewer 3 Report

Comments and Suggestions for Authors

Well written article rich in detailed information. It discusses the topic from a wide range, perhaps the part about PBPK models is a bit narrow.

A minor critical note is that the transporter localization information in the first table is a bit incomplete, it could be supplemented, e.g. BCRP also occurs on the canalicular side of hepatocytes and in the kidney. MATE-1 also occurs in the liver, P-gp is also present in the kidney, etc.

The article is suitable for publication in its current form with the correction of the above-mentioned table. I leave it to the judgment of the authors whether to supplement the paper with some additional details based on my following remarks. In one place it is written "However, in vitro transporter inhibition assays can be notorious for reporting a large false positive rate. In one instance, the false positive rate for OATPB1/3 inhibition was up to 33%". I consider this statement to be somewhat misleading, the extent to which it gives a false positive result strongly depends on the type of in vitro assay and the type of transporter. It would be interesting if they provided information on this separately for the discussed transporters, but if this is too lengthy, then the "large" indicator should be refined. It could be mentioned that the categorical separation of substrate and inhibitor in in vitro assays can cause interpretation problems. I miss the fact that MRP multidrug transporters are not mentioned in the article, which, although they cause fewer clinical problems, still have an additional role. In chapter 7, mention could also be made of the effect of chemotherapy and other long-term drug regimens on transporter expression. There is quite extensive literature on this.

Author Response

Comment 1: Well written article rich in detailed information. It discusses the topic from a wide range, perhaps the part about PBPK models is a bit narrow.

Response 1: Since our paper was intended as a short communication and because PBPK modeling is a discipline on its own, we only wanted to highlight the role of modeling of DDIs and endogenous biomarkers, assuming experts in the field of PBPK might cover this topic more extensively.

Comment 2: A minor critical note is that the transporter localization information in the first table is a bit incomplete, it could be supplemented, e.g. BCRP also occurs on the canalicular side of hepatocytes and in the kidney. MATE-1 also occurs in the liver, P-gp is also present in the kidney, etc.

Response 2: Thank you for your astute comment, we updated the table to include all major expression profiles as listed in the UniProt.org database.

Comment 3: The article is suitable for publication in its current form with the correction of the above-mentioned table. I leave it to the judgment of the authors whether to supplement the paper with some additional details based on my following remarks.

Response 3: We appreciate the Review’s suggestions and have addressed these below. We find these additions have significantly improved the manuscript.

Comment 4: In one place it is written "However, in vitro transporter inhibition assays can be notorious for reporting a large false positive rate. In one instance, the false positive rate for OATPB1/3 inhibition was up to 33%". I consider this statement to be somewhat misleading, the extent to which it gives a false positive result strongly depends on the type of in vitro assay and the type of transporter. It would be interesting if they provided information on this separately for the discussed transporters, but if this is too lengthy, then the "large" indicator should be refined. It could be mentioned that the categorical separation of substrate and inhibitor in in vitro assays can cause interpretation problems.

Response 4: Thank you for your suggestion, we have revised text to state the following (line 56-60): “However, the categorical separation between effects of study drugs as substrates or their inhibitory effects in in vitro transporter assays can, in some cases, be difficult to interpret. For instance, the false positive rate of OATPB1/3 inhibition in in vitro assays for predicting the clinical findings has been reported to be up to 33%.”

Comment 5: I miss the fact that MRP multidrug transporters are not mentioned in the article, which, although they cause fewer clinical problems, still have an additional role. In chapter 7, mention could also be made of the effect of chemotherapy and other long-term drug regimens on transporter expression. There is quite extensive literature on this.

Response 5: This is an excellent suggestion, we adapted the manuscript to include a discussion on the role of multidrug resistance (MRD) transporters in chemotherapy failure as well new approaches to address this issue (see line 327-336). In addition, we further describe potential DDIs with MRP transporters (see line 374-382).